# Graphene-Incorporated Natural Fiber Polymer Composites: A First Overview

**DOI:** 10.3390/polym12071601

**Published:** 2020-07-18

**Authors:** Fernanda Santos da Luz, Fabio da Costa Garcia Filho, Maria Teresa Gómez del-Río, Lucio Fabio Cassiano Nascimento, Wagner Anacleto Pinheiro, Sergio Neves Monteiro

**Affiliations:** 1Military Institute of Engineering—IME, Materials Science Program, Praça General Tibúrcio 80, Urca, Rio de Janeiro 22290-270, Brazil; fsl.santos@gmail.com (F.S.d.L.); lucio@ime.eb.br (L.F.C.N.); anacleto@ime.eb.br (W.A.P.); 2Department of Mechanical and Aerospace Engineering, University of California San Diego—UCSD, La Jolla, CA 92093-0411, USA or fdacostagarciafilho@eng.ucsd.edu (F.d.C.G.F.); mariateresa.gomez@urjc.es (M.T.G.d.-R.); 3DIMME, Grupo de Durabilidad e Integridad Mecánica de Materiales Estructurales, Universidad Rey Juan Carlos, C/Tulipán, s/n, 28933 Móstoles, Madrid, Spain

**Keywords:** graphene, natural fiber composite, mechanical behavior, thermal analysis, ballistic armor

## Abstract

A novel class of graphene-based materials incorporated into natural lignocellulosic fiber (NLF) polymer composites is surging since 2011. The present overview is the first attempt to compile achievements regarding this novel class of composites both in terms of technical and scientific researches as well as development of innovative products. A brief description of the graphene nature and its recent isolation from graphite is initially presented together with the processing of its main derivatives. In particular, graphene-based materials, such as nanographene (NG), exfoliated graphene/graphite nanoplatelet (GNP), graphene oxide (GO) and reduced graphene oxide (rGO), as well as other carbon-based nanomaterials, such as carbon nanotube (CNT), are effectively being incorporated into NLF composites. Their disclosed superior mechanical, thermal, electrical, and ballistic properties are discussed in specific publications. Interfacial shear strength of 575 MPa and tensile strength of 379 MPa were attained in 1 wt % GO-jute fiber and 0.75 wt % jute fiber, respectively, epoxy composites. Moreover, a Young’s modulus of 44.4 GPa was reported for 0.75 wt % GO-jute fiber composite. An important point of interest concerning this incorporation is the fact that the amphiphilic character of graphene allows a better way to enhance the interfacial adhesion between hydrophilic NLF and hydrophobic polymer matrix. As indicated in this overview, two basic incorporation strategies have so far been adopted. In the first, NG, GNP, GO, rGO and CNT are used as hybrid filler together with NLF to reinforce polymer composites. The second one starts with GO or rGO as a coating to functionalize molecular bonding with NLF, which is then added into a polymeric matrix. Both strategies are contributing to develop innovative products for energy storage, drug release, biosensor, functional electronic clothes, medical implants, and armor for ballistic protection. As such, this first overview intends to provide a critical assessment of a surging class of composite materials and unveil successful development associated with graphene incorporated NLF polymer composites.

## 1. The Birth of a Novel Class of Composite Materials

Naturally found composite materials like wood, bone, antler, and others, have been used by our ancestors since the beginning of mankind. Artificially fabricated composite materials, with chemically dissimilar phases separated by a distinct and well-connected interface, can be traced back to earlier concretes developed 7 millennia ago [1]. Documented research works on these artificial composites (now referred as the “composite materials”) began to be registered as far as 1918 [2]. However, it was only in the 1960’s that a surge in publications related to “composite materials” took place, as illustrated in Figure 1a from Scopus database [3]. Today, around 11–12 thousand papers are being annually published. Within these papers, a continuous exponential increase has occurred since 1966 in relation to “polymer composites” as also presented in Figure 1a. Indeed, this class of composites emerged as innovative materials that could be developed and adapted for specialized applications, to obtain light and high-performance materials.

Among the polymer composites, those incorporated with fibers have been considered the most important owing to their highest strength and/or stiffness on a weight basis [4]. According to Scopus [3], the first publication on “polymer composites and fiber”, in which aramid was one of the mentioned fibers, occurred in 1968 [5]. An exponential growth is also associated with this special group of fiber reinforced polymer composites as shown in Figure 1a. In particular, during the 1960’s, several high-performance synthetic fibers such as aramid, ultra-high molecular weight polyethylene, UHMWPE, and poly(p-fenileno-2,6-benzobisoxazolo), PBO, were developed and have been applied as reinforcement in polymer composites [6]. However, with the world’s population increase, which is demanding sustainable actions to prevent pollution and shortage of non-renewable resources, the replacement of synthetic materials by natural ones becomes an issue to be seriously considered. In this context, the trend of replacing synthetic fibers by natural fibers has expanded in several industrial sectors [7,8,9,10,11]. This trend is justified by desirable characteristics of natural fibers, such as abundance, recyclability, biodegradability, and bio-renewability [9,12,13]. Natural fibers, especially those of vegetable origin, also known as natural lignocellulosic fibers (NLFs) due to the predominance of cellulose and lignin in their structure, have attracted increasing attention mainly as reinforcement of polymeric matrix composites [14,15,16,17,18,19,20].

Today’s interest for “natural fiber composites” became evident by the exponential growth in publications, starting in the last decade of the 20th century, as revealed in Figure 1b [3]. More than one thousand papers are expected to be published in 2020. In fact, since the past few decades, these composites are rapidly replacing conventional composites reinforced with synthetic fibers in several technological applications, especially in the civil construction and automobile industries [7,10,11]. A recent application, which has been investigated, is the use of NLFs composites for ballistic personal protection as part of a multilayer armor [21,22,23,24,25,26,27,28,29]. In addition, composites with cellulose extracted from natural fibers have been shown efficient action to remove toxic substances from the water, such as heavy metal ions [30,31,32].

In spite of the aforementioned advantages combined with low density and cost, NLFs exhibit heterogeneity in properties, which may limit their applications. This fluctuation in properties are associated with natural variations on the chemical composition, diameter, crystalline fraction of the cellulose, fiber source type (root, stem, leaf, seed, fruit, among others), in addition to the growth stage and cultivation conditions [33,34]. Another aspect that differentiates NLFs from synthetic fibers is the hydrophilic nature of the former, which has hydroxyl groups in their composition and present moisture content between 5 and 10% [35,36]. By contrast, polymers commonly used as matrices have a hydrophobic character, which results in a weak interface with hydrophilic NFLs. For this reason, pretreatments for physical or chemical surface modification might be carried out in NFLs to improve interfacial adhesion [13,36,37]. Pretreatment can also increase the surface roughness, which contributes to adhesion on the polymeric matrix. However, some NLFs already present considerable natural roughness and surface wax. In this case, chemical pretreatment can degrade the fiber and impair the matrix reinforcement [13]. To improve the interfacial adhesion between the natural fibers and the polymeric matrix, several treatments are being extensively investigated. Among these, alkali, silane, benzoyl, acetylation, acrylation, radiation, permanganate, peroxide, and sodium chlorite have been proposed [8,9,38,39]. A recent review of the effect of these different chemical treatments on the properties of natural fiber composites was presented by Godara [38]. Kalia et al. [39] also reviewed these treatments on NLFs and concluded that most of them decrease the fiber strength.

Recently, research works using graphene incorporated materials suggested a possible emerging solution to the NLFs lower adhesion to polymer matrices [40,41,42,43,44,45,46,47,48,49,50,51,52,53,54,55,56,57,58,59,60,61,62,63,64,65,66,67,68,69,70,71,72,73,74,75,76]. This motivated, since 2015, an exponential rise in the number of publications on “natural fiber and composite and graphene” [3] given by the corresponding (red squares) curve in Figure 1b. Based on its mathematical adjustment, one may predict 190 publications on this subject in 2020. If this rise follows similar tendency of the other curves in Figure 1a,b, then a novel class of composite materials is surging. Improvements on the mechanical [50], thermal [40], and ballistic performance [51] due to the enhancement of interfacial adhesion caused by graphene oxide (GO) coating were reported. GO is important, regarding fiber coating for application in polymeric composites, owing to the functional groups present in its structure. These groups give an amphiphilic character, improving the resistance of the interface between the polymeric matrix and the NLF [51,52,69]. In addition, the presence of the negative charge in GO stabilizes the dispersion and allows the interaction with the functional groups of the fibers/fabrics [62,77].

## 2. Graphene-Based Materials

Graphene is a flat monolayer of carbon atoms in a two-dimensional (2D) honeycomb lattice, which is formed by strong sp2-carbon bonds. This is the basic unit for graphitic materials of all other dimensions, such as fullerenes with zero-dimension (0D), nanotubes (1D), and graphite (3D) [78,79,80,81]. In the literature, it is common to find the expression “graphene” referring to its derivatives or graphene-based materials (GBMs), which often causes misunderstandings by the reader. A detailed description of all graphene derivatives is beyond the scope of this overview; only the main variations of graphene used in composites will be mentioned here. The reader may find a detailed nomenclature of GBMs in the work by Bianco et al. [79].

The first report about this “lamellar structure of graphite” was in 1859 by Brodie [82]. In the work entitled “On Atomic Weight of Graphite”, the author described the oxidation and washing process of the graphite that resulted in a light-yellow substance, which is now known as graphene oxide (GO). This “carbonic acid”, as referred by Brodie [82], was also thermal reduced, resulting in another popular derivative from graphene, now named rGO. However, it was only in 2004, that two physicists, Geim and Novoselov, successfully isolated the graphene from graphite using the “Scotch Tape” method. This discovery led to a Nobel Prize for Physics in 2010, resulting in massive growth in research on this new material [80]. More than 4350 publications (articles and patents) on graphene were reported in the following year, against only 40 publications before 2004 [83]. This growth has continued, reaching the mark of 20,000 graphene publications in 2019 [3]. In a worldwide ranking, China and USA account for almost 60% of publications related to graphene, followed by South Korea and India with 8 and 7%, respectively [3].

Graphene has exceptional properties such as extensive surface area (2630 m^2^∙g^−1^) [84], high Young’s modulus (~1.0 TPa), high tensile strength (~130 GPa) [85], high thermal conductivity (~5000 W∙m^−1^∙K^−1^) [86], and good electrical conductivity. All these remarkable features reveal the great potential of graphene for pioneering applications, such as transparent conductive electrodes, sensors, innovative medical applications, and novel composite materials [84]. However, these high values are for graphene with no or low-defect levels [80].

The final properties of the graphene-based material depend on the number of graphene layers, average lateral dimension, and atomic carbon/oxygen ratio [87]. Among these three parameters, the number of layers is widely used to define some of the structure’s derivatives from graphene. A single layer of bonded sp2-hybridized carbon atoms is known as monolayer graphene or pristine graphene (PG), while graphene with 2–5 layers is known as few-layer graphene (FLG). The multilayer graphene (MLG) has 5–10 layers [79]. For optical and electronic applications, the number of layers is extremely important. For example, the optical absorption of graphene, as well as its electrical resistance, is proportional to the number of layers. Graphene with 1 to 5 layers (FLG) reflects only <0.1% of the incident light in the visible region. This rises to ~2% for 10 layers (MLG), while the electrical resistance decreases with the increase in the number of layers [88].

The structure, properties, and the number of layers of the graphene-based materials are closely related to the method chosen for their production [78]. These processes can be divided into two main categories: the top-down and the bottom-up approaches [89]. The top-down approach brings together several manufacturing methods, which aim to break another structure into atomic layers, such as mechanical exfoliation, graphite intercalation, cutting of carbon nanotubes (CNT), reduction of graphite oxide, electrochemical exfoliation, among others. The bottom-up approach consists of methods using an alternative source of carbon as a building block. The growth of metal-carbon castings, epitaxial growth in silicon carbide (SiC), and chemical vapor deposition (CVD) are examples of the bottom-up approach. Each technique results in different properties that might be suitable for specific applications. Many of them involve a high cost, which makes its application impractical [89].

The most cost-effective production method is by liquid-phase exfoliation, producing dispersions of graphene [90,91]. This technique was formerly developed by Hummers and Offeman in 1958 [92], to obtain the graphite oxide by the oxidation of graphite using a mixture of permanganate and sulfuric acid. Since then, the so-called modified or improved Hummers’ methods to obtain graphene oxide have been significantly optimized [93,94,95]. Marcano et al. [93] eliminated NaNO_3_, increased the amount of KMnO_4_ and performed the oxidation reaction in a 9:1 mixture of H_2_SO_4_/H_3_PO_4_, instead of only H_2_SO_4_, improving the efficiency of the oxidation process and eliminating toxic gas evolution. Chen et al. [95] proposed a method without using NaNO_3_, avoiding the release of toxic nitrogen oxides gases. The GO could remain exfoliated in water and later to be reduced back to graphene. For this reason, this graphene-based material has been successful in several applications, such as electronics, conductive films, electrode materials and composites [84,95,96,97]. This method is now the most applied to functionalize NLF polymer composites, as further discussed.

## 3. Composites with Graphene-Based Materials

This section presents the research efforts in NLF/polymer composites incorporated with graphene. The use of graphene-incorporated materials in these composites is discussed as well as their related properties. In addition, a brief description of the main techniques to produce these composites is presented.

The graphene discovery led to the emergence of new opportunities for the development of novel high-performance and lightweight composites, enabling its application in several fields, from electronics to aerospace. Indeed, a low filler content of this material or its derivatives can significantly improve mechanical, thermal, and physical properties [70,98,99,100,101,102,103,104,105,106]. As previously mentioned, the final properties of graphene-based polymer composites can be affected by many factors, including the type of graphene used, its dispersion state in a polymer matrix, its interfacial interactions and the method used to produce the composite [99]. Several manufacturing techniques have been used to produce the graphene-based composites. Among them, the melt blending, solution mixing, in situ polymerization, and coating are the most common [100,107]. These techniques and the main graphene components used in polymer composites are schematically illustrated in Figure 2.

Among the processes shown in Figure 2, the melt blending is more commercially attractive due to its practicality and versatility, particularly for thermoplastic polymers. Moreover, this technique may be considered environmentally friendly because is free of toxic solvent [100]. However, it involves shear force, which may impair the graphene dispersion into the polymer matrix owing to viscosity increasing at higher filler contents in the composite [107].

In the in situ polymerization, the graphene-incorporated material is mixed with a liquid monomer and an initiator. The reaction is controlled by temperature and time and can also be initiated by radiation [100,107]. In this method, well-dispersed graphene in a polymer matrix is produced. However, the increase in viscosity during the reaction limits the load content. Thus, in some cases, a solvent is added to prevent this shortcoming. On the other hand, its removal is a critical issue. The same problem occurs in the solution mixing process, which consists of the dispersion and incorporation of the filler in a solvent, followed by solvent evaporation [107]. This is the most practical technique, as it allows the easy dispersion of GO in a polar organic solvent, water, for example. Moreover, it makes more homogeneous composites in comparison to the other two methods already mentioned [100].

To improve the fiber’s properties, the coating method is usually applied. It can be done by either immersion into a graphene-based dispersion or spraying the graphene dispersion directly onto the fiber. The first method, the most common, is also known as “dip and dry” and can be followed by a reduction or washing process [100]. The coating method has been widely applied to electronic textiles because it promotes electrical conductivity, without impairing their flexibility and stretchability [41]. These novel materials are discussed in Section 5. Table 1 presents the manufacturing processes for incorporation of graphene-based materials into natural and synthetic fibers and their composites using different methods, such as exfoliated graphene/graphite nanoplatelets (GNPs), graphene flakes (G), graphene oxide (GO), reduced graphene oxide (rGO), nanographene (NG), and carbon nanotube (CNT).

Du et al. [112] investigated the incorporation of GO in glass fiber/epoxy composites by solution mixing and obtained an 18% increase in tensile strength for the composite with 15 wt % of glass fiber and 5 wt % of GO. An even higher tensile strength improvement was achieved by Fan et al. [111], who reported an increase of 85% for polymethyl methacrylate (PMMA) composite produced by melt blending and reinforced with 0.7 wt % aramid fiber coated with GO. For natural fiber-based composites, this improvement was also observed by Chen et al. [53]. In their work, polypropylene (PP) modified with maleic anhydride grafted polypropylene (MAPP) was used as matrix composite reinforced with 7.5 wt % sisal fiber coated with GO. This composite exhibited a tensile strength of 86% higher than pure PP, as the MAPP enhanced the compatibility between fiber and polymer matrix by 15%. In the paper by Sarker et al. [50], the addition of 0.75 wt % of GO into jute/epoxy composite resulted in a tensile strength two times greater than the composite with untreated jute fibers. In addition, Saker et al. [52] also observed a notable increase (~96%) in a single jute fiber coated with 1 wt % of GO, as well as an increase in interfacial shear strength by ~236%.

The improvement in the mechanical properties of polymeric materials with the addition of graphene-based materials was reviewed by Papageorgiou et al. [114]. They concluded that stiffness and strength of polymers with low Young’s modulus, such as elastomers, display higher increases in these parameters than thermoplastics or thermosetting polymers. On the other hand, the Young’s modulus of graphene incorporated in composite materials was higher for stiffer thermoplastic or thermosetting matrices with low filler content. This behavior can be noted for different graphene-based materials, such as GO and rGO, as shown in Figure 3 [114].

Many works have reported the effect on the mechanical [50,53,65,71,72,108,109,110,111,112,113,115], thermal [51,53,65,100,115] and electrical [50,52,115,116] properties of fiber composites reinforced with graphene-based materials. A detailed discussion of NLFs-GO/polymer matrix composites properties can be found in Section 4.

Table 2 presents the mechanical properties of natural fiber (NLF)-polymer composites, NLF composites with graphene, and NLFs/graphene materials used in Ashby plot shown in Figure 4. This figure reveals the superior mechanical properties of the graphene-based natural fiber polymer composites.

In Figure 4, the Ashby plot presents a comparison of Young’s modulus and tensile strength of natural fiber-based composites (orange), natural lignocellulosic fibers (NLFs) with GBMs (green), and their composites (pink) against synthetic fibers composites (glass and carbon), and graphene-incorporated polymer composites reported in the review by Kinloch et al. [81]. In this plot, the mechanical properties of carbon fibers, carbon nanotubes (CNT), graphene materials, and other common polymeric materials are included. For natural fiber-polymer composites, NLF/graphene-based composites, and NLF/graphene material, the mechanical properties were based on many reported values from the literature, as shown in Table 2.

The good relation between stiffness and strength of NLF/graphene-based composites can be noted in Figure 4, in comparison to the synthetic glass fiber composites (GFRP). In this figure, together with results presented in Figure 2, it is established the superiority of graphene-based natural fibers composites in terms of the Young’s modulus and tensile strength. It is also worth mentioning that the manufacturing process of both graphene and composite might have a significant effect on the mechanical properties, especially in the case of composites with unidirectional fibers. An approach to this effect in NLF-based composites, without graphene, was reviewed by Shah [17].

A great effort is being made to develop natural fiber composites with graphene materials as reinforcement since they exhibit outstanding mechanical properties. In general, the composites are produced using graphene either as a filler, which is mixed directly in the matrix, or coating the fiber, which is used as reinforcement. A potential progress is disclosed in both cases for low volume fractions of graphene-based materials and is presented as follows.

## 4. Properties of Graphene Incorporated NLF Composites

Since the rise of graphene in 2004, the application of graphene and its derivatives became a big hit in the materials science field. Researchers from different areas started to investigate the incorporation of graphene (G), graphene oxide (GO) and reduced graphene oxide (rGO) in all kind of materials. Such investigations disclosed the influence on mechanical and electrical as well as thermal and optical properties that could be improved in comparison to “neat” materials. It did not take long to this approach be considered in high performance composite materials [73,139]. As indicated in Section 3, two major strategies for the incorporation of graphene in NLFs polymer composites were adopted, which included its use either as filler or coating/functionalization constituent. In the following sections it will be discussed the use of graphene in polymer matrix composites reinforced with NLFs under these two main strategies and how their incorporation affects different properties. In Section 4.1 the use of graphene as filler in NLF polymer composites is evaluated in terms of mechanical and thermal properties. In Section 4.2 the GO-coating of NLFs and their incorporation in polymer composites is assessed by means of mechanical and thermal properties as well as ballistic resistance and other characteristics. It is important to notice that a special attention will be given to the functionalization of NLFs with GO. For better understanding the other cases, especially those using synthetic fibers, the readers might refer to the following works [100,140].

### 4.1. Graphene as a Filler in Natural Fiber Polymer Composites

#### 4.1.1. Mechanical Properties

The first strategy was to use GO as filler in polymer matrix composites associated with some fiber reinforcement. The successful development of these hybrid composite structures lies on the synergy between the matrix and reinforcements as well as in the optimized composition of the constituents. Prasob et al. [49] investigated the mechanical behavior of a hybrid composite using jute fiber and rGO filler reinforcing epoxy matrix. Figure 5 presents the results for different mechanical properties measured in three different temperatures. One may notice that using graphene and it is derivatives, an increase is observed in the mechanical properties of NLFs reinforced polymeric matrix composites. However, the increment reached might be considered low and in many cases the amount of GO could negatively impact in the properties of the composite. In fact, the amount of filler is something that should be carefully addressed, as additions in the order of 1 wt % could lead to the agglomeration of the nanofillers.

Chaharmahali et al. [71] investigated the best combination of incorporation of nano graphene oxide scattered into bagasse flour reinforced polypropylene composites. Compositions combining bagasse flour reinforcement from 15 to 30 wt % and nano-GO from 0.1 up to 1 wt % were produced and compared to samples without the addition of GO (control samples). The authors indicated that the combination of 0.1 wt % of GO and 30 wt % of bagasse flour was able to reach superior values of tensile, ~30%, and flexural, ~7%, strength while notched Izod impact was slightly decreased comparing to control samples. It was also showed that higher contents of nano-GO, 0.5 or 1 wt %, have a negative influence on the mechanical properties as the GO was easily agglomerated during the processing of the composite.

Similar results have been reported by Sheshmani et al. [73], who evaluated the properties of GNP/wood flour/PP composites. In their paper, the wood flour was extracted from *Populus deltoides* and the fraction of 20 wt % was kept constant for all composites. The degradation temperatures were shifted to higher values for all GNP loadings, but the optimal composition of 0.8 wt % exhibited the highest improvement in the thermal stability and lower water absorption. Although the incorporation of GNP into the PP matrix improves mechanical properties, the SEM micrographs showed that high contents (>3 wt %) of GNP were easily agglomerated, as illustrated in Figure 6, decreasing the mechanical properties [73].

Another nanofiller tested in natural fiber-based composites is the carbon nanotube (CNT), which is considered as a graphene sheet rolled up forming 1D structure. However, difficult dispersion of CNT in the polymer matrix and high cost make it not as attractive as other graphene derivatives. Mechanical and fracture properties of ramie fiber reinforced epoxy composites with filler loadings 0.2, 0.4, and 0.6 wt % of CNT were evaluated by Shen et al. [141]. The results showed an improvement in flexural and viscoelastic properties by increasing CNT content; though the impact toughness decreased. These are similar results obtained by Wulan et al. [142] for CNT/fruit bunch palm oil fiber/epoxy composite.

Lima et al. [143] reported on the effects of combining curaua fibers and GO nanometric particles in polyester matrix composite. Using Taguchi method, the authors produced different composites with the quantity of nanometric particles of GO varying from 0.1 up to 1 wt % and curaua fiber reinforcement from 10 to 30 wt %. Tensile and flexural strength were reported with exceptional 156% and 186% increase, respectively, in comparison to the neat polyester. Unlike the Chaharmahali’s work [71], Lima et al. [143] concluded that the increasing the amount of GO could be beneficial for improving the tensile and flexural strengths of the composite. However, when compared to the tensile strength of the neat curaua/polyester composite [144], no improvement was noted as the 1 wt % GO-filler curaua/polyester composite reached a tensile strength of 49 ± 1 MPa while the neat curaua/polyester composite achieved 61 ± 15 MPa. Indeed, by considering the standard deviation these results are the same. One should be wondering why it was not observed an increment of the composite properties. The answer is probably associated with processing parameters such as the proper scattering of the nano-GO into the composite. Figure 7 shows the presence of agglomerates nanoparticles in bagasse flour/PP composites with 1 wt % of GO filler [71]. The presence of the agglomerates results in the generation of flaws and subsequent creation of voids between the filler and the matrix polymer. This may directly impair the mechanical properties of the material.

#### 4.1.2. Thermal Properties

Idumah and Hassan [67] studied the effect of GNP on thermal properties of kenaf fiber/PP composites produced by melt blending. They tested different contents of GNP, from 1 to 5%, as well as the influence of maleic anhydride grafted PP (MAPP) on the interfacial adhesion and GNP dispersion into the PP matrix. The authors observed a higher thermal conductivity with an increase in GNP loading. This strong dependence was justified by the superior thermal conductivity (3000 W/mK) and high aspect ratio of GNP, which promotes interconnected network structures resulting in greater thermal conduction for the composite. However, the GNP restricts chain mobility reducing the crystallinity of PP in their composites and shifting the degradation to higher temperature. In other words, chain restrictions led to a higher thermal stability. Although the GNP acted as a barrier to volatile products during thermal decomposition, the highest thermal stability was exhibited by the composite with 3% of GNP attributed to its more homogenous dispersion [67].

A variant of the first strategy in the use of graphene or its derivatives, to produce composites with superior resistance, could be the functionalization of the polymer matrix. This approach is commonly used for producing composites consisting of polymeric matrix modified by GO functionalization. Layek et al. [145] reported several techniques aiming the graphene chemical functionalization during polymers synthesis. Although polymer-GO functionalization, based on covalent and non-covalent bonding, is well reported, few researches investigated this technique in association with reinforcement of NLFs. Costa et al. [146] investigated GO-functionalized two-component epoxy resin, during the polymerization, to be reinforced with long and aligned curaua fiber. They showed that the functionalization of the epoxy matrix increased the yield and tensile strength in 64 and 40%, respectively, in comparison to the untreated composite. Tshai et al. [65] investigated the effect of graphene functionalization in epoxy matrix composite reinforced with palm empty fruit bunch fiber. Graphene, both pristine as-produced (PG) and commercially available (CG), with amounts varying from 0.01–0.05 wt % were added into the epoxy resin and the fiber reinforcement was set as 18 wt %. Results indicated that the addition of 0.01 wt % of PG into composites was associated with maximum, tensile and flexural properties values, exhibiting the highest Young’s modulus and higher thermal stability. However, more compatibility of the CG was observed for 0.05 wt % of PG. The morphological study also showed that high content of PG (0.05 wt %) was easily agglomerated, which resulted in a decrease of ~40% in both tensile and flexural strength. This higher content of PG did not significantly affect the thermal properties of composites.

In another investigation, Han et al. [72] explored the functionalization strategy between nanomaterials such as GNP and micro-size reinforcements such as kenaf natural fibers, in polylatic acid (PLA)-based composites. The first process step was coating kenaf fiber with GNP using the conventional “dip and dry” method. Then, the GNP coated kenaf fibers were mixed with the PLA pellets using a twin-screw extruder at 180 °C. Finally, the composite samples were produced by injection molding at 75 °C. Three-point bending test and DMA were carried out to evaluate the flexural and viscoelastic properties of composites with 20, 30, and 40 wt % of kenaf fiber and 1, 3, and 5 wt % of GNP. The flexural strength and modulus and the viscoelastic properties such as storage modulus were determined. The authors showed that addition of 5 wt % GNP did not increase the viscosity of the polymer melt, but enhanced the flexural modulus by 25–30% at any fiber loading used. The addition, however, did not increase the strength, indicating insufficient load transfer at the interface.

Despite not using graphene nor its derivatives, but exfoliated graphite instead, the relevance of that investigation lies in it being the precursor of the functionalization strategy. The use of graphite-based material as coating for natural fiber proved to be a strategy worth of investigation, which regain researcher’s attention in the end of 2020 decade. Such strategy suggests that it could prevent fiber delamination and also delay the crack propagation by redistributing the stress around fibers [147,148]. Different natural fibers have been investigated under this strategy such as: jute [50,52], sisal [53], curaua [51,146] and piassava [40]. The effect of GO-coating approach on natural fibers for polymer matrix (PM) composites reinforcement extends way beyond of the aforementioned surface modification with enhancement of interface adhesion between the natural fiber reinforcement/polymer matrix. This kind of functionalization was also reported to provide some remarkable improvement on tensile, flexural, thermal, water resistance and ballistic impact properties of NLF-GO-PM composites as will be shown following.

### 4.2. Natural Fibers Coated with Graphene as Reinforcement of Polymer Composites

#### 4.2.1. Mechanical Properties

Table 3 presents the main reported mechanical results for works on NLFs polymer composites incorporated with graphene.

Sarker et al. [50] discussed the development of ultra-high-performance composites by nano-engineered GO-coating of jute natural fiber. Their procedure displayed remarkable enhancement of mechanical properties as the tensile strength and elastic modulus were improved over ~110% and ~320%, respectively, in comparison to the composites with untreated jute fiber composites. These amazing enhanced properties were obtained by a combination of physical and chemical treatments.

The enhanced mechanical properties of GO treated jute/epoxy composites were associated with two main reasons:
Strong adhesion between the GO and alkali treated fiber; andInteraction of GO treated jute fiber with the matrix. The oxygen containing functional groups present in GO could create strong bond with the alkali treated fibers to make them capable of carrying more load from the matrix.

The authors also observed by SEM fractographic analysis that the untreated fiber failure mechanism is predominantly associated with weak interfacial bonding and fiber pull-out. As for the GO-treated jute fibers, a modification on the failure mode occurred from fiber pull-out or de-bonding to transverse fracture. This indicates an improvement in the interfacial bonding that can lead to higher tensile properties of the composites.

Chen et al. [53] went further and investigated not only tensile but also impact resistance properties of sisal fiber treated with GO in a polypropylene (PP) composite. The authors discussed that the enhancement of mechanical properties would be achieved due to the introduction of GO onto the surface of the sisal fiber because it would act as a barrier between impurities, such as wax, pectin and hemicelluloses, and the PP matrix. Simultaneously, numerous hydroxyl and carboxyl groups of GO sheets on sisal fiber (SF) could not only react with the hydroxyl groups of the natural fiber but also improve the wettability between it and PP matrix.

With the increasing of fiber content, the tensile strength, the tensile modulus and the impact strength of the PP composites increased up to 7.5 wt % and then turned down. Higher fiber loading (10 wt %), however, promotes stress concentration and induced more defects, which led to a weak interface adhesion between the fibers and the PP matrix. Meanwhile, the elongation at break significantly decreased with the increase of fiber loading, which was associated with the restriction of molecular chain mobility of PP. In summary, the GO-SF/MAPP-PP composite with 7.5 wt % fiber loading had the highest increase of 36.50, 30.00, 69.73, and 36.27% in tensile strength, tensile modulus, elongation at break and the impact strength comparing to the neat PP.

The surface fracture of the tensile specimens of both the treated and untreated composites are presented in Figure 8. One may notice that voids and cracks are observed in Figure 8a due to the debonding of the sisal fiber. In Figure 8b narrower cracks between the fiber reinforcement and PP matrix are observed. This behavior was attributed to the increased surface roughness and the enhanced mechanical interlocking between SF and PP matrix [53,149]. It was observed in Figure 8c that incorporating MAPP into the PP composites improved the interface adhesion. The combined treatment of GO and MAPP, showed in Figure 8d, enhances the interface adhesion between SF and PP matrix effectively, which is assigned to chemical reaction of GO and MAPP.

It is clear that, to effectively improve the mechanical strength of NLF-GO-PM composites, a previous treatment of the surface is important, which would allow that impurities inherent to natural fibers such as hemicelluloses or waxes be removed. With that, a stronger bonding between the natural fiber reinforcement and the GO-coating would improve the transfer of loads from the matrix to the reinforcement.

#### 4.2.2. Thermal Properties

Garcia Filho et al. [40] used the modified Hummers-Offeman method to produce a 0.56 mg/mL GO solution and coat the piassava fiber prior to their incorporation into the epoxy matrix. The authors analyzed the piassava fiber by thermogravimetric analysis (TGA)/derivative thermogravimetry (DTG) and reported that the peaks of the three main stages of degradation had shifted from 64.2, 288 and 359 °C in the untreated piassava to 62.3, 317 and 479 °C, respectively, for the GO-piassava fiber. The first peak presented a slight decrease, which was attributed to the moisture content of the fiber itself and could be an indicative that the GO coating might decrease the water absorption of the fiber, as will be further discuss. The two other peaks are related to the maximum degradation rate of hemicellulose and lignin, respectively. Therefore, one may infer a clear thermal protection effect of the GO coating on the piassava fiber, as its main constituents (hemicellulose and lignin) maximum degradation rate is shifted to significantly higher temperatures. Similar investigation was carried out for GO-treated and untreated curaua fiber [51]. In the differential thermal analysis (DTA) for the untreated curaua fibers, three stages were observed: the first one between 250 and 300 °C, referring to the degradation of the hemicellulose, with maximum rate at 272 °C. The second been carried out around 293 and 350 °C, maximum rate at 327 °C, which was associated with cellulose decomposition. In the third stage the lignin decomposition took place, in temperatures around 400 and 450 °C and maximum rate on 422 °C. However, a different behavior was presented by the GO-treated curaua fibers. Their maximum degradation rate was shifted to higher temperatures and this effect may indicate an increase in the thermal stability of the fibers as discussed for the piassava fiber [40]. The effect may be due to the formation of an insulation to heat propagation by the GO coating, which retarded degradation and improved thermal stability.

Dynamic mechanical analysis (DMA) was used to investigate how GO-coated piassava epoxy composites behave at different temperatures for different amount of fiber reinforcement [40]. DMA results revealed remarkable changes caused by the amount of fiber reinforcement in the epoxy matrix composites as well as the functionalization of such fibers by the GO coating. Figure 9 presents the dynamic modulus of the investigated conditions.

The dynamic modulus, also known as storage modulus, is often associated with the stiffness of the material. It measures the elastic or potential energy stored by the material [150]. It was discussed that, despite the similar shape of the curves displayed by both cases, neat and GO-coated, three different regions are observed. There are also some observed differences, which were associated with the GO-coating of the piassava fiber. The first difference lies in the measured values for the dynamic modulus (E’). The epoxy matrix composites reinforced with neat piassava, Figure 9a, displays higher values of E’ in comparison with those reinforced with GO-coated piassava fiber, Figure 9b. The GO coating increased the stiffness of the material, thus it would result in lower values of E’. In addition, an inverse trend related to the amount of reinforcement was observed in both conditions. In the non-coated fiber condition, higher amounts of reinforcement shift the dynamic modulus for higher values. On the other hand, the GO-coated fiber reinforcement higher amounts bring E’ for lower values. GO as a functionalized graphene presents groups such as hydroxyl and carboxyl obtained by oxidation process. These groups could improve the interaction between fiber and polymer matrix, leading to a higher glass transition temperature. Because of high interface strength, both relaxation process and storage modulus of GO composites decreases slower. For the same loading fraction, graphene incorporated composites exhibit higher interface strength between filler and matrix, which requires higher activation energy for the glass transition, resulting in better thermal stability of polymer composites [81]. This may be an indication how the load is transferred from the matrix to the reinforcement. Thus, indicating the quality of the adhesion between fiber and matrix.

Sisal fiber (SF)/polypropylene (PP) composites were investigated by TGA and differentials scanning calorimeter (DSC) [53]. The influence of the GO-coating of the sisal fiber as well as the co-functionalization of the PP matrix with MAPP was shown to impair the thermal resistance of these composites. Figure 10 presents corresponding TGA and DSC curves.

It is possible to verify from Figure 10a that, in comparison to the SF/PP, the initial and maximum decomposition temperatures are improved by GO treatment and combined treatment of GO and MAPP. This enhancement of thermal stability may indicate the better interface adhesion between the SF and the PP matrix. Figure 10b displays the DSC heating curves of SF/PP, GO-SF/PP, and GO-SF/MAPP-PP composites. It is observed that the melt peak temperatures of SF/PP, GO-SF/PP, and GO-SF/MAPP-PP composites are at 172, 173, and 177 °C, respectively. These phenomena might be associated with the effective increment of the crystallinity of the composite due to the improved interface adhesion between SF and the PP matrix.

Damage from heat may be considered one of the major drawbacks for the use of natural fibers reinforced polymeric composites in engineering applications [151]. Because of that, several researchers directed their efforts on the thermal assessment of NLFs and their polymer matrix composites [149,150,151,152,153,154,155,156]. However, improvement of thermal resistance has been considered somehow difficult to reach by traditional surface treatment of NLFs such as alkali, anhydride and silanation. Nevertheless, Premkumar et al. [157] showed that the basalt fiber hybridization could shift the onset of thermal degradation of curaua fiber to 50 °C, in comparison to untreated curaua fiber. The authors also reveal that GO-coating strategy provided a higher enhancement on the thermal resistance and stability of NLFs and their composites.

#### 4.2.3. Ballistic Performance

Before considering the use of graphene-base materials, the application of NLFs as reinforcement of polymer matrix composites for ballistic protection was investigated [22,23,24,25,26,27,28,29,158,159,160,161,162]. The ballistic performance of these composites was found to be in the same order of magnitude of commonly used materials such as Kevlar^TM^ and Twaron^TM^, which are aramid-based fibers, or Dyneema^TM^ and Spectra^TM^, produced with UHMWPE. The ballistic performance obtained in NLFs polymer composites was shown to be a function of different energy absorption mechanisms, Figure 11, that includes capture of fragments, fibrils separation, fiber pullout, composite delamination, fiber breaking, and matrix rupture.

Recently, the possible effect of graphene-based material on the ballistic performance of NLF polymer composites was investigated. Target results are illustrated in Figure 12. Costa et al. [51] produced curaua fiber (CF) reinforced epoxy composites without, Figure 12a and with, Figure 12c, GO-coated curaua fibers. These composites were ballistic tested in accordance with NIJ 0101.04 standard [163] using 7.62 mm high energy ammunition. The authors reported that in none of the tested conditions the composite target was perforated by the ballistic impact. Indeed, the GO-coated condition (GOCF) caused an indentation on a clay witness, simulating a human body, less than the NIJ standard limit of 44 mm. The GO-coating was associated with the enhancement of the interfacial adhesion between the curaua fiber and the epoxy matrix, resulting in the composite target, Figure 12d, being able to keep its physical integrity, unlike the CF condition, Figure 12b, wherein a catastrophic failure was observed.

The improved interface between fiber and matrix might reduce the amount of absorbed ballistic energy by some of the aforementioned mechanisms. Nevertheless, the better physical integrity is a required condition for ballistic protection vests. summarizes the comparison of the GOCF conditions studied by Costa et al. [51] and several other untreated NLFs polymer composites with the same amount of fiber reinforcement subjected to NIJ 0101.04 [163] ballistic test.

The advantage regarding the strategy of GO-incorporation in NLFs polymer composites for ballistic protection application can be seen in Table 4. One should notice that all reported conditions exhibited penetration depth in the range from 17 up to 28 mm, which is less than the aforementioned 44 mm limit of the NIJ standard [163]. On the other hand, it is possible to see that the physical integrity was obtained almost exclusively for the composites, in which the reinforcement architecture was based on more than one direction. The only exception was observed for the GO-incorporated curaua fiber one direction aligned epoxy composite, that due to the enhancement of the interfacial resistance between the reinforcement and matrix was able to keep its physical integrity after the ballistic impact of a high energy ammunition.

#### 4.2.4. Other Characteristics

The water absorption of sisal fiber (SF) reinforced PP composites, with or without GO, was addressed by Chen et al. [53]. Figure 13 presents the evolution of water absorption percentage of the investigated conditions up to 45 days. One may notice in this figure that the PP neat condition did not absorbed any water during the 45 days test. This behavior is consistent with the hydrophobic nature of polymeric materials. On the other hand, the water absorption percentage of PP composites exhibit a rising trend, which is associated with the inherent hydrophilic property of the sisal fiber. Moreover, as expected, the highest water absorption percentage is obtained for the composite with higher amount of fiber reinforcement. It is possible to see that the GO-SF was able to significantly decrease the water absorption of the 10 wt % sisal fiber reinforcement PP matrix composite. The authors further investigated the modification of PP matrix with MAPP associated with the GO-SF. They found that condition was responsible for the lower values of water absorption among the composites. This suggests that the functionalization of both the matrix and the fiber could lead to lower values of absorbed water in NLFs composites. In the same paper, Chen et al. [53] used X-ray diffraction (XRD) results to demonstrate that GO could play a role as nucleating agent for PP crystallization. This may be a way to rationalize the mechanism of how the enhancement between the NLFs and the polymer matrix is achieved for thermoplastic polymer matrix.

Graphene-coated NLFs can promote new functionalities and can be used as energy storage, filtration, drug release, biosensors, functional clothes, medical implants, and others. For example, in the work by Qu et al. [164], a novel ultraviolet (UV) blocker for personal protection made of a cotton fabric functionalized with low content (0.05 to 0.4 wt %) of GNP was developed and its performance evaluated. The results showed that adding 0.05 wt % of GNP, the UV transmittance of control fabric decreased 2.5 times. This decrease was even greater, up to 15 times, for the content of 0.4 wt % GNP. Besides, in the washing durability test, after 10 wash cycles, the UV transmittance barely changed, indicating a stable fabric surface coating and efficient UV blocker. 

Shateri-Khalilabad and Yazdanshenas [165] also coated cotton fabric with graphene materials. In their work, the cotton fabric was dipped into the aqueous dispersion of GO with a concentration of 0.05% and then dried at 90 °C. This process was repeated three times and the resulting fabric was immersed in different reducing agents (NaBH_4_, N_2_H_4_, C_6_H_8_O_6_, Na_2_S_2_O_4_, and NaOH). Both reducing agent type, concentration, and immersion time had a great influence on the electroconductivity of the cotton fabrics, resulting in an electrical resistivity ranging from 10^3^ to 10^6^ kΩ.cm^−1^. Among the reducing agents tested, Na_2_S_2_O_4_ achieved the highest level of electrical conductivity, and 30 min reaction was enough for the complete reduction of graphene oxide [165]. The use of cotton fabric as an electronic textile coated with graphene-based materials was also the topic of some studies conducted by Karim and Novoselov’s group [41,50,52] and are discussed in Section 5.1.

A different approach was done by Weng et al. [166], who produced a graphene-cellulose paper (GCP) composite to be used as flexible electrodes in supercapacitors. The GCP was produced by vacuum filtration of a graphene nanosheet (GNS) suspension through a filter paper. The presence of functional groups on cellulose fibers and the negative charge of graphene promoted electrostatic attraction between these components, resulting in a conductive interwoven network [166]. Similarly, in the work of Kang et al. [167] GNS was chemically dispersed in a cellulose pulp and filtered. The resulting composite was highly flexible, allowing it to be folded without damaging or peeling off the copper foil. This composite also showed high electrochemical activity as an electrode in both a supercapacitor and lithium battery with a promising electrochemical performance [167]. Detailed information of these and other cellulose-graphene composites used for flexible supercapacitors can be found in the review paper by Xing et al. [168].

Supercapacitors fabricated from graphene/cotton composites as flexible electrodes have also demonstrated good capacitance and low resistance [116,169]. In the work by Liu et al. [116], GO suspension was painted onto the cotton fabric using a brush-coating technique and dried at 150 °C for 10 min. Annealing treatment at 300 °C for 2 h in argon atmosphere was applied and the GO/cotton composite was changed to GNS/cotton composite, resulting in a strong interfacial adhesion and good flexibility [116].

It is important to notice that the development of the strategies for incorporation of graphene-based materials into NLFs polymer composites still walks in babies’ steps and many issues, especially in the processing sector, still lack investigation and should be properly addressed. Moreover, it is also clear that the benefits of the incorporation of graphene are much bigger than one could think. Indeed, several properties such as tensile, flexural, impact resistance, ballistic penetration, physical integrity of composites, water resistance, thermal resistance and stability, and many others were shown to be improved by the GO-incorporation.

## 5. Applications and Future Trends of Graphene Incorporated NLF Composites

The global market for graphene application is growing rapidly, in association with scalable and more efficient production methods that are continuously being developed. Some applications, such as electronics, semiconductors, and battery are already a reality. Indeed, cell battery based on graphene is already available on the market and can be mass-produced. This battery can charge 3 times faster and has a longer lifetime in terms of charge-cycle than lithium ones. Besides, it generates less heat [170]. As for NLF polymer composites incorporated with graphene, one may expect outstanding properties, which might enable a wide range of applications, such as flexible electronics for medical devices, sportswear, military goods, among others. Some of these applications are now discussed.

### 5.1. Electronic Textiles

Electronic textile (e-textile), including natural fabrics, allows the collection, processing, storage, and display of information. These wearable sensors are revolutionizing human-machine interaction in several applications, such as functional clothing, healthcare sensors, interior design, and military applications [171,172]. E-textiles, also known as smart textiles, have been an active topic of research recently. As reported by Stylios [173], more than USD 5.55 billion will be spent in the healthcare sector with e-textiles in the next five years, which shows the relevance of this matter.

Since the 2000s, many publications are dedicated to e-textiles [171,173,174]. However, the materials commonly used to produce them, such as silver and copper, are expensive, nonbiodegradable, and toxic [175]. Hence, new solutions for environmentally friendly e-textiles have been explored over the last years [115,171,172,175,176,177,178]. Among these studies, the e-textiles coated with graphene-based materials have already shown great promise [172,175,176,179,180,181,182,183,184,185,186,187]. According to Scopus database [3] for the title/abstract/keywords “graphene” and “e-textile”, 140 papers were published in journals in the last decade.

Natural fibers, such as cotton [172,175,176,184,185], silk [177], flax [178], and wool [115] in woven and knitted structures coated with graphene-based materials [186,187] have been investigated as e-textiles. Ren et al. [176] proposed a flexible conductive cotton fabric coated with GO by vacuum filtration followed by a hot press method to its reduction (rGO). After 10 washing cycles, this e-textile kept the low sheet resistance of ~1.2 kΩ/sq exhibiting a good response even after 400 bending cycles indicating the viability of this e-textile as a strain sensor. A novel dyeing method to produce graphene-based conductive cotton yarn integrated into a knitted fabric structure was devised by Afroj et al. [172] to enable scalable production. These authors obtained a washable, flexible e-textile with both great temperature sensitivity and cyclability, allowing the data sending via a wireless device. This wearable e-textile could enable monitoring the physiological conditions of a human body without impairing the comfort [172]. An rGO-silk fabric was investigated as a pressure sensor in the work by Liu et al. [177]. In their work, silk fabric was soaked in a GO solution and then thermal reduced, obtaining a multi-layer structure of silk/rGO. This natural device exhibited a high sensitivity pressure of 0.4 kPa^−1^ and a measurement range of up 140 kPa, which was considered a relevant performance by the authors [177]. In the work by Souri and Bhattacharyya [178], a novel graphene nanoplatelets coating method by ultrasonication bath has been proposed and applied in flax fiber yarns. The authors developed strain and pressure sensors using this conductive natural fiber. The electrical properties were evaluated through tests to monitor human motion, such as wrist, joint movements (finger and knee), respiration, and pronunciation movements. Flexible touch panels are a possible application for the pressure sensor proposed, which detects the human’s finger [178].

According to Xu et al. [115], the wearable e-textiles of natural fibers (cotton, silk, flax) coated with graphene-based materials that have been reported until now [172,175,177,178] may present long-term electrical and mechanical instability due to the temperature variations and moisture from metabolic activities of the human body. Therefore, they proposed a wearable sensor of wool-knitted fabric coated with GO followed by the L-ascorbic acid reduction process, which was capable to keep good electrical and mechanical properties with moisture up to 90% [115]. The authors indicated that, after 10 washing cycles, the sheet resistance of ~2 kΩ/sq exhibited by the wool/rGO conductive textile was higher than the compressed and encapsulated graphene-based cotton e-textile (< 1 kΩ/sq) reported in the work of Afroj et al. [41].

In the light of the aforementioned investigations, the future wearable electronics must be flexible, resilient, comfortable, and present potential to mass production and low cost. Natural fibers treated with graphene-based materials have been shown as promising candidates [41,176,177,178]. These novel solutions and discoveries have been pushing the boundaries of in graphene-incorporated natural fiber composites forward.

### 5.2. Gas Sensor

Graphene stands out as a material for nano-sensors owing to its single-atom thickness. This makes graphene extremely sensitive to changes in its environment. In addition, the presence of functional groups makes the graphene oxide highly sensitive to water molecules and can also easily react with many other chemicals. Therefore the graphene-based materials can be used for humidity and gas sensors in the detection of various hazardous gases, such as nitrogen dioxide (NO_2_), ammonia (NH_3_), hydrogen (H_2_), hydrogen sulfide (H_2_S), and carbon dioxide (CO_2_) [188].

Using these properties, fiber-based reduced graphene oxide (rGO) gas sensors were fabricated by GO coating in natural fibers textiles and yarns [188,189,190]. Yun et al. [189] developed a rGO/cotton yarn sensor weaved into a cotton fabric with an embedded LED circuit, which lights on upon exposure to 5.0 ppm of NO_2_. Three types of commercial silk fabric weaving coated with GO thermal reduced were investigated as NO_2_ sensors by Jung et al. [190]. The oxygen functional groups of GO promoted hydrogen bonding with the amide groups of silk fiber, which changed into an sp2 hybridized hexagonal carbon structure after heat treatment [190].

A simple, low cost, and environmentally friendly NO_2_ sensor was proposed by Kumar et al. [191]. Graphene layers were directly transferred onto natural paper for resistive sensing of NO_2_ with a remarkably low limit of detection at 300 ppt. First, the graphene was grown on Cu foils by a chemical vapor deposition process, then polymethyl methacrylate (PMMA) film was spin-coated forming PMMA-graphene film, which was transferred onto paper and dried at room temperature. After that, the PMMA layer was dissolved in acetone for 5 min, resulting in a graphene/paper composite (g-paper).

## 6. Final Remarks

Based on Scopus metrics, this first overview disclosed that the scientific community might be witnessing the birth of a novel class of natural fiber polymer composites, which is exponentially surging in the present decade. The main characteristic of this class is the incorporation of graphene-based materials to act as reinforcement or to improve the interfacial adhesion between hydrophilic natural fibers and hydrophobic polymer matrix.

As a polymer matrix reinforcement, different types of graphene-based: (i) pristine graphene (PG); (ii) nanographene (NG); (iii) graphene flakes (G); (iv) exfoliated graphene/graphite nanoplatelets (GNP) have effectively been incorporated for superior thermal, mechanical, machinability, and viscoelastic properties. Improvement of the interfacial adhesion is attained by coating natural fiber or fabric with graphene oxide (GO) or reduced graphene oxide (rGO). The presence of GO functional groups interacts with hydroxyl groups in the fiber cellulose chains modifying the fiber surface by providing new favorable interfacial bonds.

Indeed, among the applications associated with graphene-based materials, the incorporation in natural fiber/fabric polymer composites has very recently attracted special attention. Remarkable properties justify the rapid increase in publications and motivates the attempts to develop engineering products. Indeed, from the aforementioned exponential growth, it is predicted that around 3000 articles on graphene-incorporated natural fiber polymer composites might be published by 2025.

Graphene-based materials have been shown to significantly improve stiffness, toughness, electrical conductivity, and as well as the thermal stability of natural fiber polymer composites. Based on the articles discussed in this overview, one should expect a promising future perspective for research works on incorporation and functionalization of natural fibers with GBMs for a new generation of high-performance composites.

On the other hand, there are relevant technological challenges to be overcome, especially concerning the graphene, such as its reproducibility to large-scale applications, as well as the homogeneous dispersion in composites and its interfacial interactions. The safety, reliability, and durability of these composites are still subjected to investigations.

There is no doubt that graphene incorporated natural fiber composites should promote innovative research works and contribute to industrial developments, but more studies are needed to provide a better understanding of the interaction between components in this surging novel class of composite materials.

## Figures and Tables

**Figure 1 polymers-12-01601-f001:**
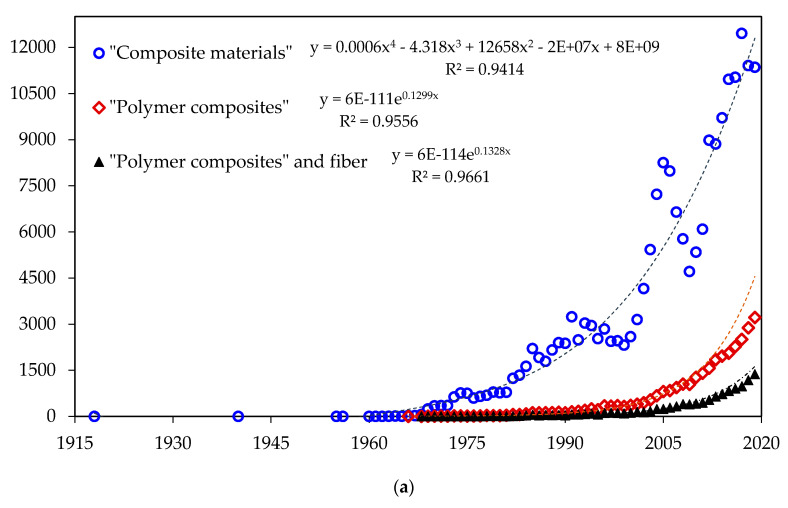
Publications by year for different material classes, according to Scopus database [3]: (**a**) “Composite materials”, “Polymer composites”, and “Polymer composites and fiber”; (**b**) “Natural fiber and composite” and “Natural fiber and composite and graphene”. In corresponding equations, y is the number of publications, and x is the year.

**Figure 2 polymers-12-01601-f002:**
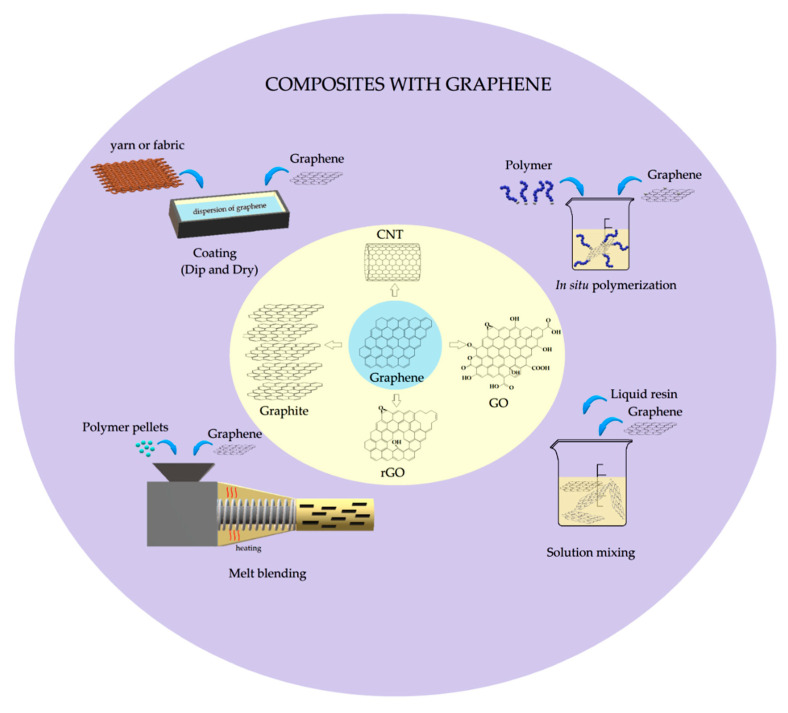
A schematic summarizing the main manufacturing processes and graphene-based materials used in composites.

**Figure 3 polymers-12-01601-f003:**
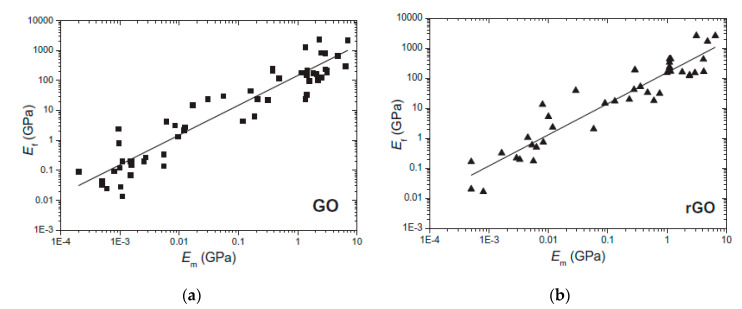
Young’s modulus (E_f_) of graphene-based materials as a function of the modulus of the matrix (E_m_): (**a**) graphene oxide (GO); (**b**) reduced graphene oxide (rGO). Adapted from [114].

**Figure 4 polymers-12-01601-f004:**
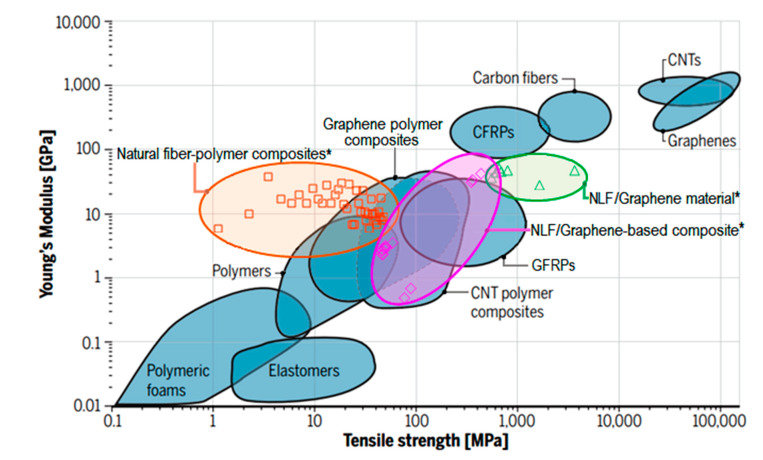
Ashby plot of Young’s modulus vs. tensile strength comparing the mechanical properties of natural fiber-based polymer composites* with glass fiber–reinforced plastic (GFRP) and carbon fiber–reinforced plastic (CFRP), with carbon nanotube (CNT), and graphene–based polymer composites. *The mechanical properties of natural fiber-polymer composites, natural lignocellulosic fiber (NLF)/graphene-based composites, and NLF/graphene material were based in many reported values from the literature, as shown in Table 2. Adapted from [81].

**Figure 5 polymers-12-01601-f005:**
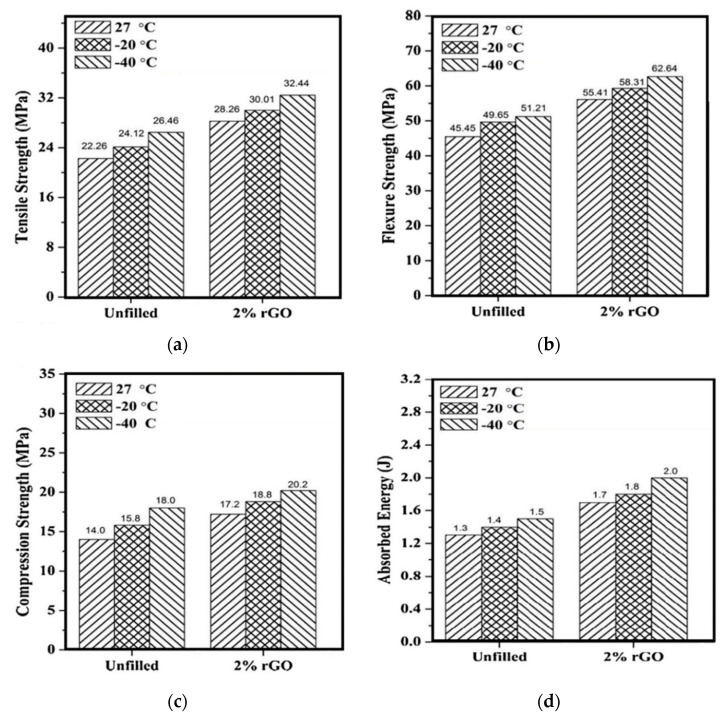
Properties of jute/epoxy composites incorporated and non-incorporated with rGO: (**a**) tensile, (**b**) flexural, (**c**) compression, and (**d**) Izod impact resistance. Adapted from [49].

**Figure 6 polymers-12-01601-f006:**
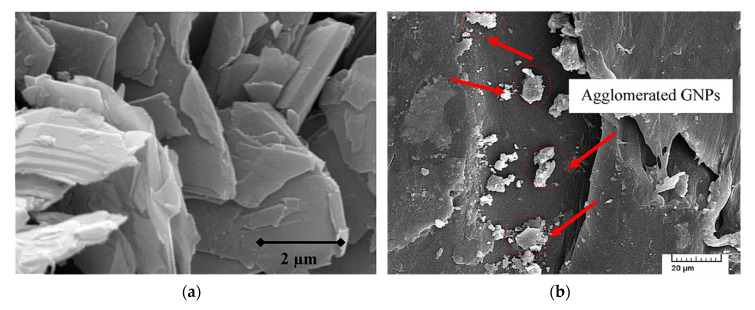
SEM microphotographs of: (**a**) pristine GNP; (**b**) composite filled with 5 wt % GNP showing agglomerated GNPs. Adapted from [73].

**Figure 7 polymers-12-01601-f007:**
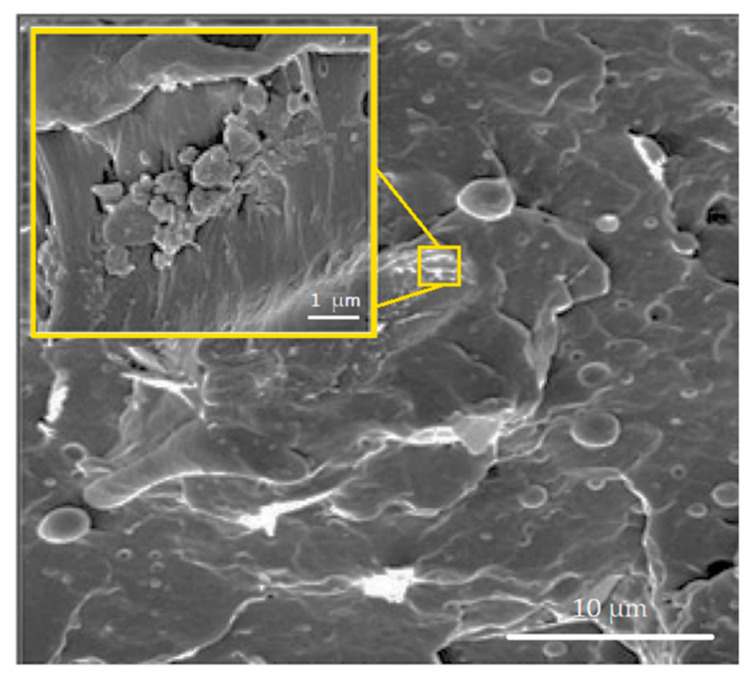
SEM micrograph of bagasse flour/PP incorporated with 1 wt % of GO filler. Inset displays the agglomeration of the filler in the fractured surface. Adapted from [71].

**Figure 8 polymers-12-01601-f008:**
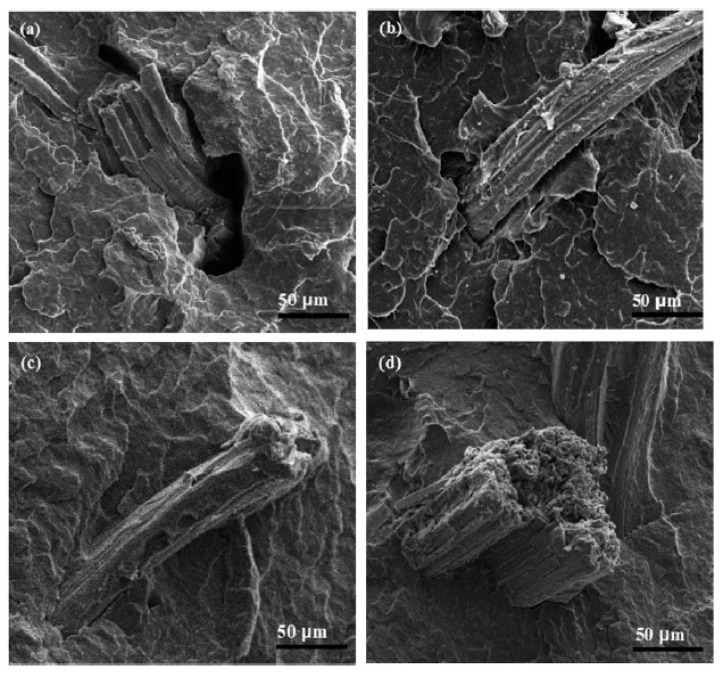
Fracture surface micrographs of PP composites: (**a**) SF/PP composite; (**b**) GO-SF/PP composite; (**c**) SF/MAPP-PP composite; (**d**) GO-SF/MAPP-PP composite. Reproduced with permission from [53].

**Figure 9 polymers-12-01601-f009:**
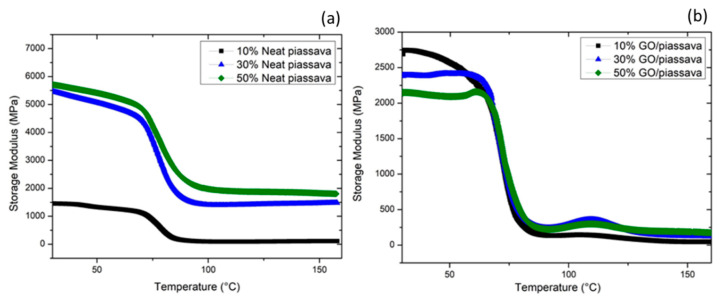
Dynamic modulus behavior for different amounts of (**a**) neat piassava fibers and (**b**) GO-coated piassava fibers. Adapted from [40].

**Figure 10 polymers-12-01601-f010:**
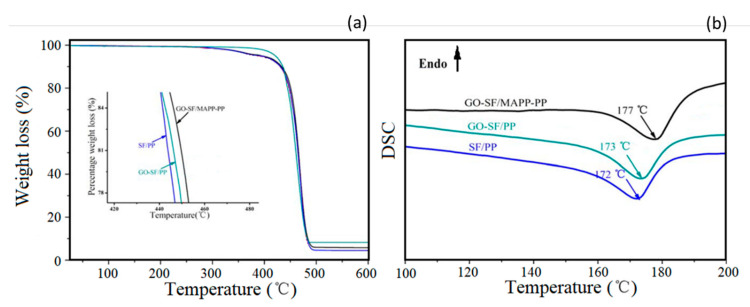
Thermal behavior of sisal/PP composite with the incorporation of GO (fiber) and MAPP (matrix): (**a**) TGA curves, and (**b**) DSC analysis. Adapted from [53].

**Figure 11 polymers-12-01601-f011:**
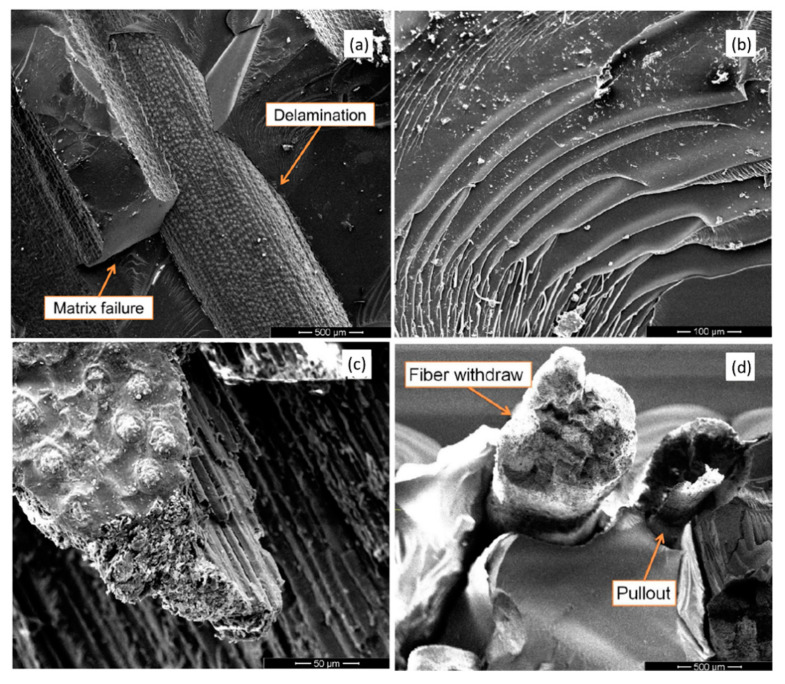
SEM micrograph displaying several micro mechanisms of failure observed in NLFs polymer composites under ballistic impact. Adapted from [158].

**Figure 12 polymers-12-01601-f012:**
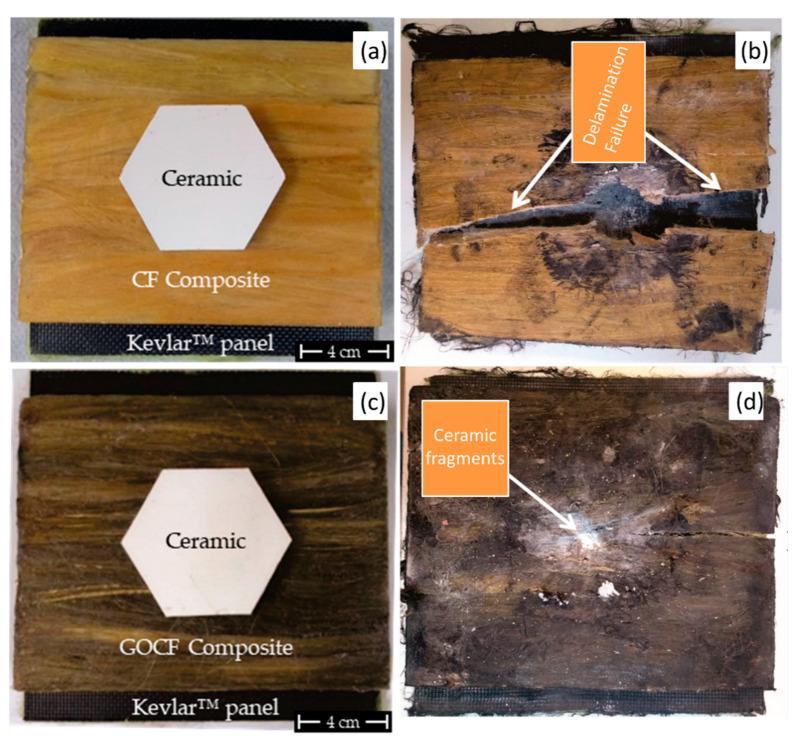
Multilayer armor system using curaua/epoxy composites as second layer: (**a**) CF before test, (**b**) CF after test, (**c**) GOCF before test, and (**d**) GOCF after test. Adapted from [51].

**Figure 13 polymers-12-01601-f013:**
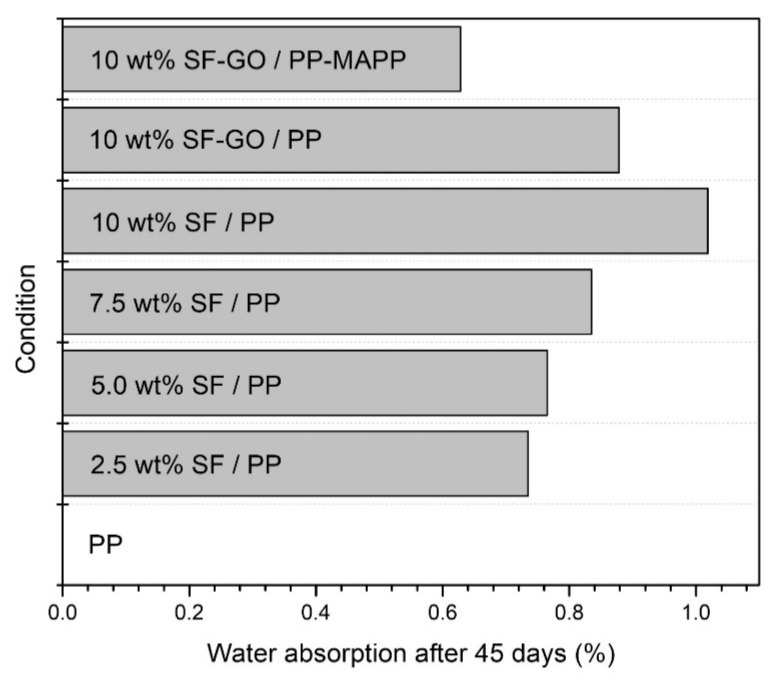
Water absorption of different sisal/PP composites after 45 days. Adapted from [53].

**Table 1 polymers-12-01601-t001:** Manufacturing processes of natural and synthetic fibers with graphene-based material, and their composites.

Material	Manufacturing Process	Reference
Natural fiber		
1 wt % GO-Curaua	Coating	[51]
Jute	Coating	[52]
0.25 wt % GO-Jute		
0.5 wt % GO-Jute		
0.75 wt % GO-Jute		
1 wt % GO-Jute		
1 wt % G-Jute		
10 wt % G-Jute		
Natural fiber-based composite		
Piassava/epoxy	Fiber: Coating; Composite: Compression molding	[40]
GO-10 vol% Piassava/epoxy		
GO-30 vol% Piassava/epoxy		
GO-50 vol% Piassava/epoxy		
Jute/epoxy	Fiber: Coating; Composite: Vacuum infusion	[50]
0.25 wt % GO-Jute/epoxy		
0.5 wt % GO-Jute/epoxy		
0.75 wt % GO-Jute/epoxy		
1 wt % GO-Jute/epoxy		
PP	Fiber: Coating; Composite: Melt blending	[53]
7.5 wt % Sisal/PP		
GO-7.5 wt % Sisal/PP		
GO-7.5 wt % Sisal/MAPP-PP		
15 wt % bagasse flour-0.1 wt % NG/PP	Melt blending	[71]
15 wt % bagasse flour-0.25 wt % NG/PP		
15 wt % bagasse flour-0.5 wt % NG/PP		
15 wt % bagasse flour-0.75 wt % NG/PP		
15 wt % bagasse flour-1 wt % NG/PP		
30 wt % bagasse flour-0.1 wt % NG/PP		
30 wt % bagasse flour-0.25 wt % NG/PP		
30 wt % bagasse flour-0.5 wt % NG/PP		
30 wt % bagasse flour-0.75 wt % NG/PP		
30 wt % bagasse flour-1 wt % NG/PP		
PLA	Fiber: Coating; Composite: Melt blending	[72]
5 wt % GNP/PLA		
40 wt % Kenaf/PLA		
40 wt % Kenaf-1 wt % GNP/PLA		
40 wt % Kenaf-3 wt % GNP/PLA		
40 wt % Kenaf-5 wt % GNP/PLA		
Synthetic fiber		
Aramid fiber	Coating	[108]
GO (ph6)-Aramid fiber		
GO (ph9)-Aramid fiber		
Glass fiber	Coating	[109]
0.4 mg/mL CNT-glass fiber		
1 mg/mL CNT-glass fiber		
1.6 mg/mL CNT-glass fiber		
2 mg/mL CNT-glass fiber		
Synthetic fiber-based composite		
Epoxy	Solution mixing	[110]
5 wt % GNP/Epoxy		
10 wt % Glass fiber-5 wt % GNP/Epoxy		
15 wt % Glass fiber-5 wt % GNP/Epoxy		
PMMA	Solution mixing	[111]
0.3 wt % rGO/PMMA		
1 wt % Aramid fiber/PMMA		
0.7 wt % Aramid fiber-GO/PMMA		
Glass fiber/PES	Fiber: Coating; Composite: Melt blending	[112]
0.1 wt % GO-Glass fiber/PES		
0.2 wt % GO-Glass fiber/PES		
0.5 wt % GO-Glass fiber/PES		
1 wt % GO-Glass fiber/PES		
PP	Fiber: Coating; Composite: Melt blending	[113]
10 wt % Glass fiber/PP		
1 wt % GNP-10 wt % Glass fiber/PP		
3 wt % GNP-10 wt % Glass fiber/PP		
5 wt % GNP-10 wt % Glass fiber/PP		
7 wt % GNP-10 wt % Glass fiber/PP		

**Table 2 polymers-12-01601-t002:** Mechanical properties of natural fibers with graphene-based material, and their composites, used in Ashby plot.

Material	Tensile Strength (MPa)	Young’s Modulus (GPa)	Reference
Natural Fiber/Polymer composite			
~50 wt % Harakeke/Epoxy	223	17	[117]
72 wt % Flax (yarn)/PP	321	29	[118]
73 wt % Sisal (alkali treated)/Epoxy	410	6	[119]
77 wt % Sisal (aligned)/Epoxy	330	10	
30 wt % Hemp (carded)/PLA	83	11	[120]
~46 wt % Flax (sliver)/Epoxy	200	17	[121]
30 wt % Hemp (aligned)/PLA	77	10	[122]
30 wt % Flax (yarn)/PP	89	7	[123]
35 wt % Jute (woven)/UP	50	8	[124]
39 wt % Flax (aligned)/PP	212	23	[125]
44 wt % Flax (sliver)/PP	146	15	[126]
46 wt % Hemp (aligned)/PP	127	11	
45 wt % Hemp/PLA	62	7	[127]
45 wt % Flax (yarn)/Epoxy	133	28	[128]
37 wt % Flax (yarn)/Epoxy	132	15	
~58 wt % Flax (sliver)/UP	304	30	[129]
46 wt % Flax (aligned)/Epoxy	280	39	[130]
50 wt % Flax (woven)/Epoxy	104	10	[131]
40 wt % Kenaf (aligned)/PLA	82	8	[132]
40 wt % Kenaf (aligned)/PHB	70	6	
52 wt % Harakeke/Epoxy	211	15	[133]
48 wt % Sisal (aligned)/Epoxy	211	20	[134]
37 wt % Sisal (aligned)/Epoxy	183	15	
37 wt % Flax (aligned)/Epoxy	132	15	
65 wt % Hemp (aligned)/Epoxy	165	17	[135]
65 wt % Hemp (DSF)/Epoxy	113	18	
~31 wt % Flax (yarn)/Epoxy	160	15	[136]
~28 wt % Flax/Epoxy	182	20	
~24 wt % Flax (yarn)/VE	248	24	
~34 wt % Flax (yarn)/UP	143	14	
50 wt % Flax (aligned)/PP	40	7	[137]
~80 wt % Kenaf (selected)/PLA	223	23	[138]
45 wt % Harakeke/Epoxy	136	11	[15]
50 wt % Hemp/Epoxy	105	9	
30 wt % Harakeke/PLA	102	8	
25 wt % Hemp/PLA	87	9	
NLF/Graphene material			
0.25 wt % GO-Jute	394	37	[52]
0.5 wt % GO-Jute	436	44	
0.75 wt % GO-Jute	501	46	
1 wt % GO-Jute	575	48	
1 wt % G-Jute	380	44	
10 wt % G-Jute	474	52	
1 wt % GO-Curaua	1834	38	[51]
Graphene/NLF/Polymer composite			
7.5 wt % Sisal/PP	55	0.5	[53]
GO-7.5 wt % Sisal/PP	60	0.6	
GO-7.5 wt % Sisal/MAPP-PP	69	0.7	
0.25 wt % GO-Jute/epoxy	295	36.9	[50]
0.5 wt % GO-Jute/epoxy	337	42.8	
0.75 wt % GO-Jute/epoxy	379	44.6	
1 wt % GO-Jute/epoxy	292.7	37.8	
1 wt % G-Jute/epoxy	290	35.8	
10 wt % G-Jute/epoxy	294	38.1	
15 wt % bagasse flour-0.1 wt % NG/PP	41	3.1	[71]
15 wt % bagasse flour-0.25 wt % NG/PP	37	2.4	
15 wt % bagasse flour-0.5 wt % NG/PP	38	2.4	
15 wt % bagasse flour-0.75 wt % NG/PP	37	2.6	
15 wt % bagasse flour-1 wt % NG/PP	37	2.7	
30 wt % bagasse flour-0.1 wt % NG/PP	47	3.6	
30 wt % bagasse flour-0.25 wt % NG/PP	41	3	
30 wt % bagasse flour-0.5 wt % NG/PP	40	3.2	
30 wt % bagasse flour-0.75 wt % NG/PP	39	3	
30 wt % bagasse flour-1 wt % NG/PP	39	3	
40 wt % Kenaf-1 wt % GNP/PLA	106	~7.5	[72]
40 wt % Kenaf-3 wt % GNP/PLA	114	~7.6	
40 wt % Kenaf-5 wt % GNP/PLA	109	~8.8	

**Table 3 polymers-12-01601-t003:** Mechanical properties of natural fibers with graphene-based material, and their composites.

Material	Property	Reference
Interfacial Shear Strength (MPa)		
Jute	295	[52]
0.25 wt % GO-Jute	394	
0.5 wt % GO-Jute	436	
0.75 wt % GO-Jute	501	
1 wt % GO-Jute	575	
1 wt % G-Jute	380	
10 wt % G-Jute	474	
Tensile strength (MPa)		
Jute/epoxy	180	
0.25 wt % GO-Jute/epoxy	295	
0.5 wt % GO-Jute/epoxy	337	[50]
0.75 wt % GO-Jute/epoxy	379	
1 wt % GO-Jute/epoxy	293	
2.5 wt % Sisal/PP	50.4	[53]
5 wt % Sisal/PP	53.6	
7.5 wt % Sisal/PP	55.2	
10 wt % Sisal/PP	52.9	
GO-2.5 wt % Sisal/PP	54.7	
GO-5 wt % Sisal/PP	57.4	
GO-7.5 wt % Sisal/PP	59.8	
GO-10 wt % Sisal/PP	55.6	
GO-2.5 wt % Sisal/MAPP-PP	61.2	
GO-5 wt % Sisal/MAPP-PP	67.6	
GO-7.5 wt % Sisal/MAPP-PP	69.1	
GO-10 wt % Sisal/MAPP-PP	63.6	
15 wt % bagasse flour-0.1 wt % NG/PP	41	[71]
15 wt % bagasse flour-0.25 wt % NG/PP	37	
15 wt % bagasse flour-0.5 wt % NG/PP	38	
15 wt % bagasse flour-0.75 wt % NG/PP	37	
15 wt % bagasse flour-1 wt % NG/PP	37	
30 wt % bagasse flour-0.1 wt % NG/PP	47	
30 wt % bagasse flour-0.25 wt % NG/PP	41	
30 wt % bagasse flour-0.5 wt % NG/PP	40	
30 wt % bagasse flour-0.75 wt % NG/PP	39	
30 wt % bagasse flour-1 wt % NG/PP	39	
PLA	102	[72]
5 wt % GNP/PLA	102	
40 wt % Kenaf/PLA	97	
40 wt % Kenaf-1 wt % GNP/PLA	106	
40 wt % Kenaf-3 wt % GNP/PLA	114	
40 wt % Kenaf-5 wt % GNP/PLA	109	
Young’s Modulus (GPa)		
Jute	10	[50]
0.25 wt % GO-Jute	36.9	
0.5 wt % GO-Jute	42.8	
0.75 wt % GO-Jute	44.4	
1 wt % GO-Jute	37.8	
1 wt % G-Jute	35.8	
10 wt % G-Jute	37.9	
2.5 wt % Sisal/PP	0.51	[53]
5 wt % Sisal/PP	0.57	
7.5 wt % Sisal/PP	0.65	
10 wt % Sisal/PP	0.55	
GO-2.5 wt % Sisal/PP	0.53	
GO-5 wt % Sisal/PP	0.61	
GO-7.5 wt % Sisal/PP	0.68	
GO-10 wt % Sisal/PP	0.58	
GO-2.5 wt % Sisal/MAPP-PP	0.56	
GO-5 wt % Sisal/MAPP-PP	0.66	
GO-7.5 wt % Sisal/MAPP-PP	0.73	
GO-10 wt % Sisal/MAPP-PP	0.64	

**Table 4 polymers-12-01601-t004:** Ballistic assessment of different NLFs/polymer composites.

Composite	Reinforcement Architecture	Penetration Depth (mm)	Physical Integrity	Reference
GO-incorporated composite				
GO-Curaua/epoxy	Long aligned fibers	27.4 ± 0.3	Yes	[51]
Untreated composites				
Jute/polyester	Non-woven fabric	24 ± 7	Yes	[24]
Mallow/epoxy	Long aligned fibers	22 ± 1	No	[25]
Sisal/polyester	Long aligned fibers	22 ± 3	No	[28]
Piassava/epoxy	Long aligned fibers	20 ± 4	No	[29]
Curaua/epoxy	Long aligned fibers	25.6 ± 0.2	No	[51]
Fique/polyester	Long aligned fibers	17 ± 2	No	[159]
Fique/epoxy	Bidimensional fabric	22 ± 2	Yes	[161]
Curaua/epoxy	Non-woven fabric	28 ± 3	Yes	[162]

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
