# Peer review of "Graphene-Incorporated Natural Fiber Polymer Composites: A First Overview"

_polymers, 2020, doi:10.3390/polym12071601_

Round 1

Reviewer 1 Report

The manuscript entitled “Graphene-incorporated natural fiber polymer composites: A first overview” compressively reviews the recent advances in the natural fiber polymer composites enhanced by graphene. The manuscript was well organized and written and the topic is aligned with the themes of Polymers and interesting for its readers. Therefore, the reviewer would like to recommend it for publication after minor revisions.

Comments:

  1. Line 202, “(All parts of this figure were drawn by the authors)” can be removed.
  2. Line 605. Figure 12 was drawn based on the published results. Therefore, the reference is mandatory for the caption.
  3. Lines 765-772. “Author Contributions” should be written according to the Journal instruction.
  4. Lines 773-776. Please edit the Funding information!
  5. Some of the abbreviations have been identified several times. For example, graphite nanoplatelet (GNP) in Lines 22 and 364!

Author Response

The authors would like to thank the Reviewers for the valuable comments and suggestions on the structure and scientific aspects that contribute to improve the manuscript. Amendments are provided accordingly. Responses to each comment are listed below and all modifications/additions were marked as Track Changes in the revised version of the manuscript.

Reviewer #1:

Comment 1: The manuscript entitled “Graphene-incorporated natural fiber polymer composites: A first overview” compressively reviews the recent advances in the natural fiber polymer composites enhanced by graphene. The manuscript was well organized and written and the topic is aligned with the themes of Polymers and interesting for its readers. Therefore, the reviewer would like to recommend it for publication after minor revisions.

Response: The authors thank the reviewer for the complements as well as the valuable suggestions.

Comment 2: Line 202, “(All parts of this figure were drawn by the authors)” can be removed.

Response: Complied. Removed as requested. 

Comment 3: Line 605. Figure 12 was drawn based on the published results. Therefore, the reference is mandatory for the caption.

Response: The reviewer is correct. The correct reference is now added to the revised version of the manuscript.

Comment 4: Lines 765-772. “Author Contributions” should be written according to the Journal instruction.

Response: Complied. The authors contribution is now written in accordance with the journal instruction.

Comment 5: Lines 773-776. Please edit the Funding information!

Response: Complied. Funding information is now correctly identified.

Comment 6: Some of the abbreviations have been identified several times. For example, graphite nanoplatelet (GNP) in Lines 22 and 364!

Response: This is now corrected in the revised version of the manuscript.

Reviewer 2 Report

Manuscript: Graphene-incorporated natural fiber polymer composites: A first overview

Polymers-862177

Manuscript presents very good review about Graphene and natural fiber composite materials and can accepted for publication after minor change.

  • Abstract should contain some quantitative information also.
  • Novelty of the work be established.
  • All the results reported be compared in a tabular form to establish the superiority of the work.
  • Authors must need to incorporate (following) recent and important references related to the preparation and application of cellulose and their composites in the manuscript to make it more interesting for the readers. For example.
  • https://doi.org/10.1016/j.carbpol.2018.12.010
  • ACS Sustainable Chem. Eng.2019, 7, 5, 5045–5056
  • Journal of Polymers and the Environmentvolume 27, pages148–157(2019)
  • https://doi.org/10.3389/fmats.2019.00235
  • ACS Sustainable Chemistry & Engineering 6 (3), 3279-3290
  • Nanomaterials 10 (4), 706
  • ACS Sustainable Chemistry & Engineering 7 (6), 6140-6151
  • Biomacromolecules 18 (8), 2333-2342
  • Improve the quality of Figure 5.
  • Authors need to add future prospective of the presented research in the conclusion part of the manuscript.

Author Response

The authors would like to thank the Reviewers for the valuable comments and suggestions on the structure and scientific aspects that contribute to improve the manuscript. Amendments are provided accordingly. Responses to each comment are listed below and all modifications/additions were marked as Track Changes in the revised version of the manuscript.

Reviewer #2:

Comment 1: Manuscript presents very good review about Graphene and natural fiber composite materials and can accepted for publication after minor change.

Response: The authors once again thank the complement of the reviewer.

Comment 2: Abstract should contain some quantitative information also.

Response: Complied. Some quantitative information regarding the use of graphene oxide in the natural fiber composites is presented in the abstract of the revised version.

Comment 3: Novelty of the work be established.

Response: Complied. In the new version of the manuscript a special attention is given to the novelty of the paper.

Comment 4: All the results reported be compared in a tabular form to establish the superiority of the work.

Response: Indeed, the authors tried to group similar properties in a new Table 2. As recommended, grouping all reported results in one Table 2 now allows the reader to have an idea of the superiority of graphene-incorporated natural fiber composites .

Comment 5: Authors must need to incorporate (following) recent and important references related to the preparation and application of cellulose and their composites in the manuscript to make it more interesting for the readers. For example.

https://doi.org/10.1016/j.carbpol.2018.12.010

ACS Sustainable Chem. Eng.2019, 7, 5, 5045–5056

Journal of Polymers and the Environment volume 27, pages148–157(2019)

https://doi.org/10.3389/fmats.2019.00235

ACS Sustainable Chemistry & Engineering 6 (3), 3279-3290

Nanomaterials 10 (4), 706

ACS Sustainable Chemistry & Engineering 7 (6), 6140-6151

Biomacromolecules 18 (8), 2333-2342

Response: Complied. The reviewer’s suggested references are now incorporated to the review paper.

Comment 6: Improve the quality of Figure 5.

Response: The quality of Figure 5 is now improved to meet the requirements for publication.

Comment 7: Authors need to add future prospective of the presented research in the conclusion part of the manuscript.

Response: Complied, a future prospective of the research is presented in the Final Remarks section.

Reviewer 3 Report

A comprehensive review on the interesting and current topic of graphene/NLF hybrid composites. Good introduction to the subject of the article and good review of the methods of production graphene based materials used in described in the article composites. An interesting comparison, on the Ashby chart, of the mechanical properties of different types of composites. Interesting presentation of the mechanical and thermal properties of graphene composites as a filler in natural fiber polymer composites and natural fibers coated with graphene as reinforcement of polymer composites. The chapter devoted to ballistic performance of described composites is also very interesting. Described future possible applications of graphene incorporated NLF composites look also very promising. Both electronic textiles and gas sensors are very innovative applications of graphene / NLF composites. So I confirm that the rated overview gives readers good and critical assessments of development and applications of those composites. In my opinion, this review article can be published in its present form.

Author Response

The authors would like to thank the Reviewers for the valuable comments and suggestions on the structure and scientific aspects that contribute to improve the manuscript. Amendments are provided accordingly. Responses to each comment are listed below and all modifications/additions were marked as Track Changes in the revised version of the manuscript.

Reviewer #3:

A comprehensive review on the interesting and current topic of graphene/NLF hybrid composites. Good introduction to the subject of the article and good review of the methods of production graphene based materials used in described in the article composites. An interesting comparison, on the Ashby chart, of the mechanical properties of different types of composites. Interesting presentation of the mechanical and thermal properties of graphene composites as a filler in natural fiber polymer composites and natural fibers coated with graphene as reinforcement of polymer composites. The chapter devoted to ballistic performance of described composites is also very interesting. Described future possible applications of graphene incorporated NLF composites look also very promising. Both electronic textiles and gas sensors are very innovative applications of graphene / NLF composites. So I confirm that the rated overview gives readers good and critical assessments of development and applications of those composites. In my opinion, this review article can be published in its present form.

Response: The authors sincerely thank the reviewer’s complements and comments.